A simple non-invasive method for measuring gross brain size in small live fish with semi-transparent heads

Näslund Joacim joacim.naslund@bioenv.gu.se
Department of Biological and Environmental Sciences, University of Gothenburg , Gothenburg , Sweden
Bentley George
Electronic publication date: 2014 Sep 23
Publication date: 2014
Volume: 2
Electronic Location ID: e586
Received 2014 May 31; Accepted 2014 Aug 29
Copyright: © 2014 Näslund
Copyright year: 2014
Copyright holder: Näslund
License: This is an open access article distributed under the terms of the Creative Commons Attribution License, which permits unrestricted use, distribution, reproduction and adaptation in any medium and for any purpose provided that it is properly attributed. For attribution, the original author(s), title, publication source (PeerJ) and either DOI or URL of the article must be cited.
License URL: https://creativecommons.org/licenses/by/4.0/

Keywords: Optic tectum, Brown trout, Brain measurement, Zebrafish, Digital photography

Funding: Helge Ax:son Johnsons stiftelse, Wilhelm och Martina Lundgrens Vetenskapsfond, The Swedish Research Council Formas 229-2009-1495 Economical support was provided from Helge Ax:son Johnsons stiftelse, Wilhelm och Martina Lundgrens Vetenskapsfond and The Swedish Research Council Formas (grant no. 229-2009-1495, to Jörgen I. Johnsson, University of Gothenburg). The funders had no role in study design, data collection and analysis, decision to publish, or preparation of the manuscript.

==============================
This paper describes a non-invasive method for estimating gross brain size in small fish with semi-transparent heads, using system camera equipment. Macro-photographs were taken from above on backlit free-swimming fish undergoing light anaesthesia. From the photographs, the width of the optic tectum was measured. This measure (TeO-measure) correlates well with the width of the optic tectum as measured from out-dissected brains in both brown trout fry and zebrafish (Pearson r > 0.90). The TeO-measure also correlates well with overall brain wet weight in brown trout fry (r = 0.90), but less well for zebrafish (r = 0.79). A non-invasive measure makes it possible to quickly assess brain size from a large number of individuals, as well as repeatedly measuring brain size of live individuals allowing calculation of brain growth.

Introduction

Many recent studies on fish have investigated gross size of the brain and various brain areas from ontogenetic or evolutionary perspectives (Ebbesson & Braithwaite, 2012; Gonda, Herczeg & Merilä, 2013). The brain measurements have been taken using several different methods, e.g., dry or wet mass (Mayer et al., 2010; Devlin et al., 2012), one-dimensional linear measurements (Marchetti & Nevitt, 2003; Park & Bell, 2010), two-dimensional dorsal brain areas (Burns & Rodd, 2008; Park, Chase & Bell, 2012), ellipsoid volume approximations based on linear measurements (Pollen et al., 2007; Fraser et al., 2012), histological sectioning (Kotrschal & Palzenberger, 1992; Kihslinger, Lema & Nevitt, 2006), geometric morphometric analyses (Park & Bell, 2010; Park, Chase & Bell, 2012), or magnetic resonance imaging (Ullmann, Cowin & Collin, 2010; Ullmann et al., 2010). All of these methods require substantial and time consuming preparation of the brains, and the fish need to be killed before brain measurement, which precludes the possibility of getting longitudinal data on individual brain growth. Longitudinal repeated measures would be beneficial to follow brain growth allocation over experimental treatments, such as starvation/refeeding regimes and environmental manipulations, or during ontogeny of different species or genotypes. For instance, some studies suggest that brain size can be affected by addition of structures in fish rearing tanks (e.g., Kihslinger, Lema & Nevitt, 2006; Näslund et al., 2012), but such studies would benefit from following individuals through ontogeny to be able to assess effects of differential growth patterns (Näslund & Johnsson, in press). Non-invasive measurements, where the fish can be kept alive after brain measurements would also be beneficial for brain size selection experiments (e.g., Kotrschal et al., 2013). One previous study has measured brain size in live guppy Poecilia reticulata Peters, 1859 neonates, using a method similar to the one presented here (the width of the optic tectum was measured from photographs taken by a digital camera attached to a dissection microscope), but the method was not validated (see Supplemental Information in Kotrschal et al., 2013).

The aims of this paper are to (i) present a method for measuring gross brain size in small live fish with semi-transparent heads, and (ii) compare the accuracy of measurements obtained from the non-invasive method with invasive brain measurements. The species chosen for this study were brown trout (in the fry stage) and zebrafish (in the young adult stage).

Materials and Methods

Subjects

Wild brown trout Salmo trutta L. fry (N = 21, size range: 25.1–32.4 mm fork length) were collected in the stream Norumsån, Sweden (58°2.596′N, 11°50.760′E), using electrofishing (200–300 V, ≈1 A, straight DC; L-600, LugAB, Luleå) in early June, 2013. After capture the trout were kept in aquaria in the laboratory. Zebrafish Danio rerio (Hamilton, 1822) (N = 15, size range: 17.6–22.9 mm standard length; adults) were bought from a pet shop and kept in aquaria. All fish were subjected to behavioural experiments prior to brain-measurements, and were thus re-used as experimental subjects. However, this fact should not influence the results of the present study. Body length, head length and eye-diameter of the fish was measured from lateral photographs of anaesthetised fish (precision 0.1 mm), using ImageJ 1.46r (http://imagej.nih.gov/ij/). Fork length was measured in brown trout, as this is the most common measure for salmonids, while standard length was measured in zebrafish, due to the fact that some individuals had elongated caudal fins. See Fig. 1 for definitions of the body, head and eye measurements.

Figure 1 Body measurements illustrated on a schematic zebrafish.

Fork length (used for brown trout): from the tip of the snout to the fork of the caudal fin (i.e., the end of the central caudal fin ray); standard length (used for zebrafish): from the tip of the snout to the end of the spine; head length: from the tip of the snout to the dorsal-most end of the gill opening; eye-diameter: dorso-lateral diameter of the eye cavity.

Non-invasive brain measurements

A digital single lens reflex (DSLR) camera (Canon EOS 40D; Canon Inc., Tokyo) equipped with a macro lens (Canon EF 100 mm f/2.8 USM Macro 1:1) was vertically mounted on a copy-stand. Underneath the camera, a LED light box (LightPad 920; Artograph Inc., Delano, MN) was placed to provide light through the fish head, enhancing contrasts. A Petri dish was placed on the light box, underneath the lens, and filled with water containing a light dose of anaesthesia (2-phenoxyethanol, 0.3–0.5 ml l−1). Within the Petri dish a scale was placed as a reference (precision 0.01 mm). Fish were then placed in the Petri dish and photographed while being lightly anaesthetized (i.e., resting at the bottom, still keeping equilibrium). For brown trout, one photograph was normally taken, as these fish tend to rest in equilibrium (i.e., with the brain perpendicular to the camera lens) on the bottom of the Petri dish. For zebrafish, which do not rest still at the bottom of the dish, two to five photographs were taken, from which the best image (on which the head was most symmetric) was selected for measurements. In a few cases, when the fish was being too deeply anaesthetised before good photographs could be taken, I removed the fish from the anaesthesia and photographed it at a later time point. The setup of the equipment is illustrated in Fig. 2.

Figure 2 The brain photography setup.

The DSLR camera (1) was mounted vertically on a vertical copy-stand (2) equipped with a sled which could be adjusted vertically (3) Underneath the lens of the camera, on a light box (4), a Petri dish was placed, containing water-dissolved anaesthesia and the fish.

The brightness and contrast of the photographs were adjusted in Adobe Photoshop CS5.1 (Adobe Systems Inc., San Jose, CA) to maximize visibility of the brain inside the head of the fish. Measurements, i.e., the visible width, of telencephalon (Tel; only for brown trout), optic tectum (TeO) and cerebellum (Cb; only for brown trout) were taken from the digital photographs (see Fig. 3) using ImageJ 1.46r. No non-invasive Tel- or Cb-measures were taken from zebrafish, as these structures had relatively low visibility through the skin and skull in this species (Fig. 3C).

Figure 3 Dorsal photographs of brown trout fry (fork length: 31.1 mm) (A), (B), and young adult zebrafish (fork length: 23.3 mm) (C), (D). (B) and (D) show measurements taken from images.

See Fig. 4 for photos of out-dissected brains. Tel, telencephalon; TeO, optic tectum; Cb, cerebellum.

Invasive brain measurements

After being photographed the fish were killed by an overdose of 2-phenoxyethanol (1 ml l−1), and preserved in 4% phosphate-buffered formaldehyde until dissection (10 months for trout, 1 week for zebrafish). When dissected, the brains were cut off where the spinal cord enters the brain stem and all nerves were cut as close to the brain as possible. The pituitary gland was removed from all brains as they were not successfully retained from all individuals when dissected. Thereafter, they were photographed using a copy-stand mounted DSLR camera (Canon EOS 40D), equipped with a super-macro lens (Canon MP-E 65 mm f/2.8 1–5x Macro; set at ca. 1.5x macro; Canon Inc.) and two external flash heads (Canon MT-24EX Macro Twin Lite Flash; Canon Inc.). The camera was connected to a laptop computer, and pictures were captured from the computer, using the live-view function of the DSLR and the EOS Utility 2.1 software (Canon Inc.). The brains were put in Petri dishes which were filled with 0.01 M phosphate buffer solution to the height of the brain, using the liquid surface to align the brains horizontally. Within the Petri dish a scale was placed as a reference (precision 0.01 mm). After photography the brains were blotted dry under light pressure from a finger and weighed (wet mass to the nearest 0.1 mg). Measurements corresponding to the non-invasive measurements (see Fig. 4) were taken from the digital photographs using ImageJ 1.46r. One zebrafish brain had the telencephalon and optic tectum damaged during handling and no measurements of these structures could be taken from the photographs; it was still possible to measure cerebellar width and mass of this brain.

Figure 4 Dorsal photographs of brown trout fry brain (A), and juvenile zebrafish brain (C), with measurements taken marked as white lines. (B) and (D) show outlines of the trout brain and the zebrafish brain, respectively.

Tel, telencephalon; TeO, optic tectum; Cb, cerebellum. Rostral parts of the brains are pointing to the left.

Analyses

Body and brain measurements were analysed using Pearson correlation analyses in GraphPad Prism 6 (GraphPad Software, Inc., La Jolla, CA). High correlation-factors would to a high degree reflect the overall size of the fish; therefore, Pearson correlation analyses were also conducted using the deviation from expected values as predicted by body length (i.e., the residuals from the linear function of body length and dependent variables (brain measurements, head length and eye-diameter)). The latter analyses show whether non-invasive and invasive measurements have corresponding deviations from expected values, as predicted from body size. For the method to be valid these correlations should be positive and statistically significant.

Brain mass was loge-transformed prior to analysis as mass measurements scale allometrically to length measurements.

Animal ethics

The experiment was conducted in accordance with Swedish law and regulations and was approved by the Ethical Committee on Animal Experiments in Gothenburg, Sweden. Brown trout were kept under ethical license number 8-2011 and zebrafish under license number 274-2011.

Results

Brown trout fry

As judged by the Pearson correlation coefficient, r, the non-invasive TeO-measure (Fig. 3B) correlated well with the invasive measure of TeO width (r = 0.93; Table 1 and Fig. 5A) and brain mass (loge transformed) (r = 0.90; Table 1 and Fig. 5B). The non-invasive Tel- and Cb-measures did not correlate as well with their corresponding brain areas, as measured from out-dissected brains (Table 1). Eye-diameter correlated equally well with overall brain mass as the TeO-measure (r = 0.90; Table 1; Fig. 5D), while head length did not (r = 0.86, Table 1 and Fig. 5C). Body fork length also correlated well with brain mass (r = 0.92; Table 1).

Figure 5 Correlations between non-invasive measurements (X-axes) and invasive brain measurements (Y-axes).

Brown trout fry: black; zebrafish: red. Pearson r-values and parameters for the regression lines are presented in Table 1.

Table 1 Correlations between non-invasive and invasive measurements used to assess brain size.

Correlation coefficient (Pearson r) and the parameters for ordinary least squares regression lines (slope and Y-intercept) are given with estimated 95% confidence intervals within brackets.

	Correlation
coefficient (r)	Slope	Y-intercept	
Brown trout fry				
TeO-measure vs. Optic tectum width	0.93 (0.84–0.97)	0.87 (0.71–1.03)	0.41 (−0.01–0.83)	
TeO-measure vs. Loge (Brain mass)	0.90 (0.76–0.96)	0.99 (0.75–1.22)	−7.49 (−8.08 to −6.89)	
Tel-measure vs. Telencephalon width	0.57 (0.18–0.80)	0.64 (0.19–1.08)	0.47 (0.00–0.93)	
Cb-measure vs. Cerebellum width	0.63 (0.27–0.83)	0.56 (0.22–0.87)	0.48 (0.12–0.85)	
Eye-diameter vs. Loge (Brain mass)	0.90 (0.77–0.96)	1.32 (1.02–1.62)	−7.49 (−8.05 to −6.90)	
Head length vs. Loge (Brain mass)	0.86 (0.69–0.94)	0.28 (0.20–0.36)	−6.75 (−7.26 to −6.24)	
Body length vs. Loge (Brain mass)	0.92 (0.82–0.97)	0.08 (0.06–0.09)	−7.20 (−7.64 to −6.75)	
Zebrafish				
TeO-measure vs. Optic tectum width	0.92 (0.75–0.97)	0.71 (0.52–0.90)	0.61 (0.18–1.05)	
TeO-measure vs. Loge (Brain mass)	0.79 (0.45–0.93)	1.64 (0.84–2.44)	−9.35 (−11.16–7.54)	
Eye-diameter vs. Loge (Brain mass)	0.72 (0.33–0.90)	1.31 (0.55–2.07)	−8.00 (−9.37 to −6.63)	
Head length vs. Loge (Brain mass)	0.70 (0.30–0.89)	0.38 (0.15–0.60)	−7.23 (−8.20 to −6.26)	
Body length vs. Loge (Brain mass)	0.60 (0.14–0.85)	0.06 (0.01–0.11)	−6.93 (−7.94 to −5.92)	

Pearson correlations of body-size-standardized variables (i.e., using the deviation from expected values based on fork length) showed that non-invasive TeO-measurement corresponded well with the invasive TeO-measurement (r = 0.75, p < 0.0001; Fig. 6A), and with the loge-transformed brain mass (r = 0.59, p = 0.0047; Fig. 6A). This shows that a fish with small TeO-width for its body size, as measured on photographs of the living fish, also have smaller than expected TeO-width, as measured from photographs of the out-dissected brain, and lower than expected brain mass for its size. Similarly, fish with smaller than expected eyes also had lower than expected brain mass, but the statistical evidence was weaker, albeit still significant (r = 0.48, p = 0.026; Fig. 6A). Deviation from expected head length did not correlate significantly with deviation from expected brain mass (r = 0.39, p = 0.080), and comparisons of deviation from expectancy for non-invasive and invasive measurements of Tel and Cb did not produce significant correlations either (Tel: r = 0.25, p = 0.28; Cb: r = 0.33, p = 0.15).

Figure 6 Correlations of deviations from expected values as predicted by for body size.

X-axes: non-invasive measurements; Y-axes: invasive brain measurements. (A) brown trout fry; (B) zebrafish. Dotted lines show 95% confidence intervals of the regression lines.

Zebrafish

The TeO-measure of zebrafish (Fig. 3D) proved to correlate well with optic tectum width measured from invasive measurements (r = 0.92; Table 1 and Fig. 5A). Compared to brown trout fry, the TeO-measure of zebrafish was less correlated with overall brain mass (loge transformed) (r = 0.79; Table 1 and Fig. 4B). Head length (Fig. 5C), eye-diameter (Fig. 5D) and body standard length had lower correlation with brain mass than the non-invasive TeO-measure (0.60 < r < 0.72; Table 1).

Correlations of body-size-standardized variables showed that non-invasive TeO-measurement corresponded well with the invasive TeO-measurement (r = 0.89, p < 0.0001; Fig. 6B), and with the loge-transformed brain mass (r = 0.67, p = 0.0078; Fig. 6B). Like in the case of trout, this shows that deviation from expected TeO-width, as measured by the non-invasive and the invasive methods, showed similar patterns. Deviation from expected eye-diameter and deviation from expected loge-transformed brain mass also correlated, but with lower statistical support (r = 0.52, p = 0.043; Fig. 6B). Deviation from expected head length did not correlate significantly with deviation from expected brain mass (r = 0.44, p = 0.098).

Discussion

The results of this study show that a non-invasive measurement of the width of the optic tectum, taken through the skull of a live fish, correlate well with the invasive measure of the same brain structure. For brown trout fry the non-invasive TeO-measure also correlated well with brain wet mass. Other non-invasive measurements which could potentially be used as brain-size proxies (eye-diameter and head length), were generally less good. Body length of the fish also predicted brain size very well. However, experimental manipulations of growth rate (e.g., in studies involving food restriction or selection/genetic modification for rapid somatic growth) may affect the relation between body length and brain size, as fish with different growth rates have been shown to differ in brain allometry (Pankhurst & Montgomery, 1994; Devlin et al., 2012).

For both brown trout fry and zebrafish, the deviation from expected brain size based on body length correlated well for non-invasive TeO-measurements and invasive TeO-measurments. The correlations between non-invasive TeO-measurements and loge-transformed brain mass, and between eye-diameter and loge-transformed brain mass were also significantly positive, with the former having higher precision than the latter for both species. Corresponding comparisons for head size vs. loge-transformed brain mass were not satisfactory (i.e., non-significant).

The correlations between absolute values of non-invasive and invasive measurements were in general a bit lower for zebrafish than for brown trout fry. The zebrafish were smaller than the trout fry, but had less transparent heads, which may explain part of the lower correlation factor. However, body measurements (standard length, eye-diameter and head length) also showed lower correlations in zebrafish than in brown trout. This indicates that the zebrafish has more individual variation in body and/or brain morphometrics than brown trout. It should be pointed out that the zebrafish were fixed in formaldehyde for a substantially shorter period than the trout, which could have contributed to variation in invasive brain measurements. The initial period of fixation is generally when the most changes are occurring in brain tissue (Weisbecker, 2012; Ridgway & Hanson, 2014).

The main advantage of the non-invasive TeO-measure is that it requires substantially less preparation and handling time than invasive methods, making large sample sizes feasible. If there is great among-individual variation in brain size in experimental groups, a large sample size of decent brain measures could be better than a few really good brain measures. The TeO-measure also makes it possible to repeatedly measure brain growth in small live fish, e.g., through ontogeny or in growth-manipulation experiments. The main disadvantages are that the TeO-measure is limited to estimating optic tectum width and whole brain size, and it only works as long as the brain is visible through the head. The latter fact restricts repeated measures into adulthood in many species where pigmentation and thickness of tissue and bone above the brain limits the visibility of the brain.

The TeO-measure could likely be useful for estimating brain size in many small fish species and juvenile stages of larger fish species. However, since the accuracy of the TeO-measure may differ among species, validation is needed before the method is applied on a new species.

General advice

When taking photographs, it is important to make sure that the reference scale is at the same height as the dorsal-most part of the fish head in the Petri dish; otherwise it will be out of focus on the images.

Dosage of anaesthesia was more critical for zebrafish than for trout. Trout have open (physostome) swim bladder (Fänge, 1983), which is quickly emptied, and large pectoral fins which they use as support when resting at the bottom of the Petri dish. Consequently, the fish is relatively stable through the sedation. Zebrafish have a closed (physoclist) swim bladder (Fänge, 1983) and only use their relatively small pelvic fins as support while resting at the bottom. This makes the zebrafish more sensitive to changes in their equilibrium during anaesthesia, and they can flip over completely relatively quickly even if not fully anaesthetized. After a fish has lost equilibrium and flipped over, it needs to recover to regain equilibrium and then be re-anaesthetized. For this reason, the concentration of anaesthesia needs to be more carefully chosen for species with closed swim bladder, as compared to species with open or no swim bladder. The efficiency of anaesthesia depends largely on the type of anaesthesia used, species and water temperature (Ross & Ross, 2008), and suitable dosage should be tested out for each experiment.

In general, the method described works only on fish where the brain is visible through the skin and skull. The more transparent the head is, the better the measurements. This makes the method suitable, primarily, for young fish. Some adult fish of small size, like zebrafish, can also be measured using the method. Most likely, there are size thresholds when the method stops working for both zebrafish and brown trout, and likely for most other candidate species. At some particular size, the visibility through the skin and skull will be obscured by the pigmentation of the skin and/or the thickness of the skull. Based on personal observations, some fish species seem unsuitable for the method, despite small size (e.g., fish with steep foreheads like some species of cichlids), while other species have good head transparency also as full grown adults (e.g., guppies).

To make the brain maximally visible through the semi-transparent head, it is advisable to keep the fish in a bright environment for a while prior to the photography, to make the coloration of the head lighter, and the outline of the brain more easily detected.

Caution should be taken when analysing fish of different ontogenetic stages, as the relative size of the optic tectum, as compared to other brain structures, may change during ontogeny (Näslund et al., 2012).

Supplemental Information

Supplemental Information 1 Dataset

Click here for additional data file.

Per Saarinen is thanked for his assistance in collecting, caring for and photographing the trout fry. Joachim Sturve and Britt Sjöqvist are thanked for caring for the zebrafish. Jörgen I. Johnsson and three reviewers provided valuable comments to earlier versions of the manuscript.

Additional Information and Declarations

Competing Interests

Author Contributions

Animal Ethics

The authors declare there are no competing interests.

Joacim Näslund conceived and designed the experiments, performed the experiments, analyzed the data, contributed reagents/materials/analysis tools, wrote the paper, prepared figures and tables, reviewed drafts of the paper.

The following information was supplied relating to ethical approvals (i.e., approving body and any reference numbers):

The experiment was conducted in accordance with Swedish law and regulations and was approved by the Ethical Committee on Animal Experiments in Gothenburg, Sweden. Brown trout were kept under ethical license number 8-2011 and zebrafish under license number 274-2011.

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
