# Peer review of "A simple non-invasive method for measuring gross brain size in small live fish with semi-transparent heads"

_PeerJ, doi:10.7717/peerj.586_

## Round 0.1 · original submission · Major Revisions

I think this is potentially a valuable contribution, but please pay close attention to the reviewers' comments.

Reviewer 1 ·

Basic reporting

- I would include the criteria for log-transforming certain variables in the analyses section

-Please review the alphabetical order of the references (line 186)

Experimental design

- Approximate age of the fish should be stated in the methods. It is stated in the figures that juvenile zebra fish were used, but is not clear from the main text. Will this methodology work for adult zebra fish?

- It can be inferred from the text that the photographs taken for non-invasive measures were not perfectly standardize and dependent on the reaction of the fish to anaesthesia. Were photographs taken with the fish in the same position? The angle can have a big influence in the posterior measures if fish slowly start to flip and were not always perfectly positioned for a photo overhead.

Validity of the findings

Sample size is not very large, but enough for the conclusions here drawn regarding the relation of invasive and non-invasive OT-measures. Larger sample size would be required to reduce confidence intervals and assess whether non-invasive OT-measures can be used as proxies of mass measures of the brain in species with lower visibility through the skull, such as the zebrafish.

Reviewer 2 ·

Basic reporting

Overall I found this to be an interesting and short methodological paper. Fishes are important model species in studies of brain evolution and plasticity, and so I anticipate that the technique described and its overall simplicity and lack of invasiveness will appeal to the wider biological community. Having said this, however, it seems that the usefulness of the technique hinges on a couple of important points:
(1) The technique can only be applied to fishes with semi-transparent heads, and (2), that the strong relationship between optic tectum width and overall brain mass is consistent among species. Regarding the first point, does the author have any idea how many fishes actually have semi-transparent heads, either as larvae/juveniles or throughout their lives? Apparently brown trout, zebrafish and newborn guppies do, but what about other small teleost species commonly used in neurobiological studies, such as goldfish, cichlids and sticklebacks? If it were possible to get at least gave some indication of how many additional species have semi-transparent heads, it would be easier to gauge how widely applicable the technique might be. Secondly, the author only presents data on two species, the rainbow trout and the zebrafish. While the correlation coefficients for the relationships between OT-measure and brain mass for both species are quite high, the correlation for brown trout (r=0.9) is higher than for zebrafish (r=0.79). This difference may reflect species-specific differences, or it may be a result of the different fixation regimes used for the trout and zebrafish brains, a point expanded upon below. The author rightly points out that the technique would need to be validated when applied to a new species, yet this study would be stronger if the author had validated their technique in a greater number of species in the first place.

Experimental design

Lines 42-43. As this is a methodological paper the author needs to better define how head length and eye diameter were determined. E.g., regarding eye diameter, is the author referring to the transverse diameter (rostral to caudal) or the axial (corneal to scleral) diameter? Measurements commonly used in fisheries do not always align with those used by comparative neurobiologists.

Lines 55 and 74. Why were different scales (0.1 mm and 0.01 mm) used for the non-invasive and invasive methods?

Lines 64-65. There is a massive difference in the amount of time the brains were stored in fixative, i.e. 10 months for the trout vs one week for the zebrafish. Brains can undergo quite large changes in size post-fixation and these changes, characterized by initial expansion followed by shrinkage, are most apparent in the first weeks especially (e.g. Weisbecker 2012, Brain Struct Funct 217:677-685; Ridgeway and Hanson, 2014, Brain Behav Evol, eprint ahead of press DOI:10.1159/000360519). At greater time periods post initial fixation, brain size tends to stabilize. The author should consider that the assessment of the zebrafish brains after only one week of fixation, when they were likely undergoing rapid changes in size, could have influenced the results, and may in part explain the lower r-value for the correlation between OT-measure and brain mass for zebrafish compared to trout.

Lines 78-79. If the telencephalon and optic tectum were destroyed, how what it possible to measure brain weight?

Validity of the findings

I believe the findings to be valid. However, the problem of the short fixation time for the zebrafish brains and the potential bias this may have introduced at least needs to be acknowledged.

Additional comments

Introduction
Line 18. Many of the studies mentioned in the introduction have investigated the size of various brain areas as well as overall brain size in fishes. Therefore I recommend adding ‘and various brain areas’ before ‘gross brain size’.

Line 28. Change ‘needs’ to ‘need’.

Line 30. In the aims why doesn’t the author mention that the method being presented will only work for fish with semi-transparent heads?

Methods
Line 36. Change ‘in stream’ to ‘the stream’.

Line 39. Where the zebrafish adults?

Results
I know OT, TE and CE refer to optic tectum, telencephalon and cerebellum, respectively, but the definitions of these abbreviations are not formally presented anywhere in the paper, and they should be. Also the conventional neuroanatomical abbreviations for these brain areas are TeO, Tel and Cb.

Discussion

Line 115. I would have thought that the other key result that should be reiterated here is that the non-invasive OT-measure strongly correlates with brain mass (at least in the brown trout) and so offers a potentially useful proxy measure for brain size.

Line 134. Repeated measures of brain size can only be made as long as the brain is visible through the skin and skull. This is an important caveat associated with this method that should not be readily overlooked.

Line 143. Change ‘dorsal’ to ‘dorsal-most’.

Line 145. In terms of general advice, won’t anaesthesia levels be species-specific irrespective of whatever species are used?

Line 155. Has the author ever tried using some kind of cradle to help support sedated fish in the correct orientation?

·

Basic reporting

This is a useful technique for increasing the sample size of comparative experimental studies, though the introduction is lacking. This method has in fact been reported on before, though in supplementary material as the author indicates, and a brief discussion of Kotrschal et al 2013 might help frame the method, especially as this is the first true validation.
The figures are useful in gauging the methods. The acronyms for measurements labeled in photos (Figure 2, 3) should be defined in the figure legend. These measurements are not described in the methods section either, so there is a bit of jumping around for the reader. Are four panes needed for figure 2? This information could be shown in two simple panels and A and C anre just B and D with no labels. Another figure could show which morphometrics were used (body length, eye diameter).

Experimental design

The experiment is useful, but could benefit from more rigor in testing more species and at varying ages/sizes. The methods are very brief and do not describe measurements taken during the non-invasive measurement, but simply refer to figure 2. A brief discussion on why these particular structures were chosen would be useful (was it simply visibility?). There is also no analysis or discussion of the measurement of individual regions over ontogeny? While the OT-measurement was useful for what is presumably one size class of individuals there was no discussion of for varying sizes and how regional allometry may limit it’s usefulness.

Validity of the findings

The limitations of the study are appropriately acknowledged, though the idea of OT-measure is not greatly supported by their results. OT-measure was not a useful indicator of anything other than OT-width for zebrafish, suggesting this technique is not appropriate for all taxa, though the author acknowledges this. Further the OT-measure in brown trout did not perform significantly better than any other proxy of brain weight (Table 1). This finding could be more rigorously supported through the analysis of additional species with varying morphologies, and over a size range of individuals to determine when and for how long this method might be useful as a proxy for measuring brain mass.

Additional comments

This method is potentially very useful for increasing sample sizes for comparative and experimental studies, but needs further validation. One of the main benefits you mentioned was the repeated measure of individuals to measure brain growth, but there was not testing of this method over a size range. Is the method only viable for newly hatched fry? At what size does illumination and visualization of the OT and other regions become impossible? This could be validated with further analysis, as it stands the scope of the study is too limited.

---

## Round 0.2 · accepted · Accept

I think you have addressed the reviewers' comments appropriately.